# Erythropoietin Nanobots: Their Feasibility for the Controlled Release of Erythropoietin and Their Neuroprotective Bioequivalence in Central Nervous System Injury

Thi Huong Le [1], Chanh Trung Nguyen [1], Kyo-in Koo [1,*,†] and Chang Ho Hwang [2,*,†]

1   Department of Electrical, Electronic and Computer Engineering, University of Ulsan, Ulsan 44610, Korea; lehuong94alhp@gmail.com (T.H.L.); nctrung1407@gmail.com (C.T.N.)
2   Department of Physical & Rehabilitation Medicine, Chungnam National University Sejong Hospital, College of Medicine, Chungnam National University, Sejong 30099, Korea
*   Correspondence: kikoo@ulsan.ac.kr (K.-i.K.); chhwang1220@cnu.ac.kr (C.H.H.); Tel.: +82-52-259-1408 (K.-i.K.); +82-44-995-4715 (C.H.H.); Fax: +82-52-259-1306 (K.-i.K.); +82-44-995-2461 (C.H.H.)
†   These authors contributed equally to this work.

**Abstract:** Background: Erythropoietin (EPO) plays important roles in neuroprotection in central nervous system injury. Due to the limited therapeutic time window and coexistence of hematopoietic/extrahematopoietic receptors displaying heterogenic and phylogenetic differences, fast, targeted delivery agents, such as nanobots, are needed. To confirm the feasibility of EPO-nanobots (ENBs) as therapeutic tools, the authors evaluated controlled EPO release from ENBs and compared the neuroprotective bioequivalence of these substances after preconditioning sonication. Methods: ENBs were manufactured by a nanospray drying technique with preconditioning sonication. SH-SY5Y neuronal cells were cotreated with thapsigargin and either EPO or ENBs before cell viability, EPO receptor activation, and endoplasmic reticulum stress-related pathway deactivation were determined over 24 h. Results: Preconditioning sonication (50–60 kHz) for 1 h increased the cumulative EPO release from the ENBs (84% versus 25% at 24 h). Between EPO and ENBs at 24 h, both neuronal cell viability (both > 65% versus 15% for thapsigargin alone) and the expression of the proapoptotic/apoptotic biomolecular markers JAK2, PDI, PERK, GRP78, ATF6, CHOP, TGF-β, and caspase-3 were nearly the same or similar. Conclusion: ENBs controlled EPO release in vitro after preconditioning sonication, leading to neuroprotection similar to that of EPO at 24 h.

**Keywords:** nanoparticles; erythropoietin; polymers; therapeutics; regeneration; central nervous system

## 1. Introduction

Erythropoietin (EPO) has both extrahematopoietic actions and hematopoietic functions. Among its extrahematopoietic activities, neuroprotection and neuroregeneration have been widely reported both in vitro and in vivo in animal models of central nervous system (CNS) injury [1–3]. Furthermore, EPO guarantees more readily available applications for patients with CNS injuries than other experimental therapeutics and has been used worldwide for several decades to treat hematologic diseases [4]. However, EPO shows a limited therapeutic time window (6–8 h after insult) in small animal models [5–8]; therefore, quick arrival of EPO at the injured CNS area is required. Additionally, considering the neuroprotective benefits of intravenous EPO injection over the first three days to patients who have suffered a stroke [9], initial EPO administration should certainly be provided as soon as possible in the clinic. However, such administration in real-world situations is very difficult due to various problems, including hospitals being located far away, traffic jams, or time delays in the hospitals. As an alternative, preemptive EPO administration before insult to prevent neuronal injury has been proposed in brain-ischemic rodent models [10], but this kind of intervention is not relevant to healthy human populations due to potential hematologic

complications such as thrombosis, polycythemia vera, and teratogenicity [11]. Moreover, phylogenetic differences and the heterogeneity of extrahematopoietic/hematopoietic EPO receptors cause limited amounts of administered EPO to reach the injured CNS area and bind EPO receptors in situ [12]. Taking into consideration the aforementioned limitations, neuroprotective trials using EPO have remained a challenge in CNS injury.

To overcome these obstacles, targeted EPO delivery has been proposed. Indirect methods, such as epicortical implantation and focused ultrasound with microbubbles [7,13], have been studied since the late 1970s and remain at the experimental level. An alternative direct method suggested by Paul Ehrlich is referred to as the "magic bullet" [14]; in this method, superparamagnetic/paramagnetic nanoparticles (MNPs) under magnetic navigation, which have emerged as promising tools in cancer models [15–17], can be used for targeted EPO delivery. The authors confirmed that fabricated alginate-encapsulated EPO-MNPs, namely, EPO-nanobots (ENBs), could achieve magnetic navigation [18]. However, alginate, which was used as the vehicle to encapsulate EPO and MNPs, is a biodegradable polymer that, similar to other polymers, requires several days to spontaneously break down and release EPO from the ENBs. Therefore, the ENBs remain unbroken even if they are quickly and selectively delivered to the injured CNS area. As a result, the ENBs cannot expose EPO receptors to EPO within the limited therapeutic time window of EPO. Notably, preemptive sonication can promote the breakdown of encapsulating biodegradable polymers and control the release of the contained materials [19]. However, cross-links in biodegradable polymers can reform upon re-exposure to physiologic fluids after sonication-induced breakdown [20,21]. Therefore, it remains unclear whether the breakdown of ENBs is sufficient to ensure the binding of EPO to EPO receptors and the subsequent neuroprotective cascade. To evaluate the feasibility of ENBs as potential therapeutic tools in CNS injury, the authors conducted an in vitro study on the controllability of EPO release by preconditioning sonication and determined the neuroprotective bioequivalence of ENBs to EPO in an endoplasmic reticulum (ER) stress-induced neuronal cell injury model.

## 2. Materials and Methods

### 2.1. Synthesis of Magnetic Nanoparticles Using the Chemical Coprecipitation Method

The chemical coprecipitation method is the most widely used method for synthesizing MNPs [22]. $Fe_3O_4$-MNPs were synthesized via the coprecipitation reaction of $Fe^{3+}$ and $Fe^{2+}$ ions with a molar ratio of 2:1 as follows:

$$2FeCl_3 + FeCl_2 + 8NaOH \rightarrow Fe_3O_4 \text{ (s)} + 4H_2O + 8NaCl$$

First, 0.02 M ferric chloride hexahydrate ($FeCl_3 \cdot 6H_2O$) (Sigma–Aldrich, St. Louis, MO, USA) and 0.01 M ferrous chloride tetrahydrate ($FeCl_2 \cdot 4H_2O$) (Sigma–Aldrich, St. Louis, MO, USA) were dissolved in 100 mL of distilled water and stirred at 500 rpm to form a homogeneous solution. Then, a 0.8 M NaOH (Kanto, Tokyo, Japan) solution was prepared in deionized water and slowly added into the ferric-ferrous solution at a rate of 1.0 mL/minute. NaOH pumping was operated under oxygen-free conditions at room temperature to reach a pH value of 10.5 [23]. This mixture was stirred continuously for 3 h. After that, the supernatant containing residual chemicals was removed, and the MNPs were collected using a strong permanent magnet. The collected MNPs were washed with distilled water until a neutral pH was obtained and then dried at 80 °C overnight. Ultimately, these MNPs were annealed at 400 °C for 2 h before long-term storage at room temperature.

### 2.2. Fabrication of Erythropoietin-Nanobots Using a Nanospray Dryer

A B-90 nanospray dryer (Büchi Labortechnik AG, Flawil, Switzerland) was utilized to fabricate the alginate-encapsulated EPO and MNPs. The nanospray comprises two main parts: a pulsating case in the spray nozzle to atomize the feeds and an electrostatic particle accumulator to gather the fabricated particles [24]. In the current study, the authors employed a microspray nozzle (4 μm) with a flow rate of 102–106 L/minute, followed by an aspirator with a flow rate of 27.3 $m^3$/minute. The relative spray rate was adjusted to

80% of the flow rate. The inlet temperature, outlet temperature, and pressure were fixed at 120 °C, 40 °C, and 28 mbar, respectively.

To prepare the alginate-EPO-MNP solution, 7.5 mg of sodium alginate (Chem-Supply, Gillman, Australia) was mixed with 150 mg of MNPs in 100 mL of deionized water. This solution was sonicated for 1 h. After that, 1 mL of solution containing 1000 IU of recombinant human EPO (rhEPO; Epoetin alfa, Sigma–Aldrich, St. Louis, MO, USA) was added, and the resulting mixture solution was stirred for 30 min. The solution was sprayed for approximately 1–2 h using a nanospray dryer. The fabricated ENBs, consisting of 7.5 mg of sodium alginate, 1000 IU of rhEPO, and 150 mg of MNPs, were harvested from the chamber using a powder scraper. The scraped ENBs were diluted to an EPO concentration of 100 IU/mL considering dryer efficiency (80%).

*2.3. Release of Erythropoietin from the Erythropoietin Nanobots Using Preconditioning Sonication*

To enhance EPO release from the ENBs, low-frequency ultrasound (50–60 kHz) was preemptively applied for 1 h, as described in Baghbani and Moztarzadeh's report, in which low-frequency ultrasound more efficiently released loaded materials from nanobots than high-frequency ultrasound [25]. EPO release was performed in phosphate-buffered saline (PBS) at 37 °C and pH 7.4. The experiment was repeated 3 times under the same conditions. To measure the amount of the released EPO, a dotMETRIC assay (G-Biosciences, St. Louis, MO, USA) was conducted at 0.5, 1, 2, 4, 6, 12, and 24 h after sonication. EPO release was calculated as follows:

$$\text{Cumulative EPO release (\%)} = \frac{Weight\ of\ drug\ at\ each\ time\ point}{Weight\ of\ loaded\ drug} \times 100$$

*2.4. Injured Neuronal Cell Model*

For comparison of the neuroprotective bioequivalence of ENBs to EPO, the SH-SY5Y cell line (a human-derived neuronal cell line most commonly applied to in vitro neurological studies [26]) was used. SH-SY5Y cells (Sigma–Aldrich, St. Louis, MO, USA) were cultured in high-glucose Dulbecco's modified Eagle's medium (DMEM) (Thermo Fisher, Waltham, MA, USA) supplemented with 1% penicillin/streptomycin and 10% fetal bovine serum (FBS) at 37 °C under 5% $CO_2$. The SH-SY5Y cells were cultured for 2 days before treatment. Two-day-cultured SH-SY5Y cells were treated with 5 μL of 1 mM thapsigargin (TG; a noncompetitive inhibitor of the sarco/endoplasmic reticulum $Ca^{2+}$ ATPase) [27,28] only to form a sham group or cotreated with 5 μL of 1 mM TG and 20 μL of 250 IU/mL EPO to form a control (TG + EPO) group. The experimental (TG + ENB) group cells were treated with 5 μL of 1 mM TG and 50 μL of an ENB (100 IU EPO/mL) solution, as described in Table 1. The amount of EPO administered in the TG + ENB group was identical to that in the TG + EPO group (5 IU). This EPO amount was determined from reports on the neuroprotective effects of EPO against apoptosis [29–32].

**Table 1.** SH-SY5Y neuronal cell treatments.

|  | Cell Only | TG | TG + EPO | TG + ENBs |
|---|---|---|---|---|
| Culture media | 10 mL | 9.995 mL | 9.975 mL | 9.945 mL |
| 1 mM/mL TG | - | 5 μL | 5 μL | 5 μL |
| 250 IU/mL EPO | - | - | 20 μL | - |
| 100 IU EPO/mL-nanobots | - | - | - | 50 μL |
| Total volume | 10 mL | 10 mL | 10 mL | 10 mL |

TG indicates thapsigargin treatment only, TG + EPO indicates thapsigargin and EPO cotreatment, and TG + ENBs indicates thapsigargin and EPO nanobot cotreatment.

*2.5. Neuronal Cell Viability*

SH-SY5Y cells ($5 \times 10^4$) were plated in a 96-well plate and incubated at 37 °C for 24 h with 100 μL of medium per well. After 24 h of culture, 20 μL of CellTiter 96 AQ$_{ueous}$

One Solution Cell Proliferation Assay solution (Promega, Madison, WI, USA) containing a tetrazolium compound (3-[4,5-dimethylthiazol-2-yl]-5-[3-carboxymethoxyphenyl]-2-[4-sulfophenyl]-2H-tetrazolium; MTS) and an electron coupling reagent (phenazine ethosulfate) was added, after which the cells were incubated at 37 °C for 3 h. The absorbance was quantified at 490 nm using a SpectraMax iD3 reader (Molecular Devices, San Jose, CA, USA). The experiment was repeated 5 times under the same conditions. Cell viability was assessed as follows:

$$\text{Cell viability } (\%) = \frac{Absorbance\ of\ sample\ at\ 490\ nm}{Absorbance\ of\ control\ at\ 490\ nm} \times 100 \tag{1}$$

For the live/dead staining assay, SH-SY5Y cells were stained in PBS containing 0.2% (2 mM) ethidium homodimer-1 in dimethyl sulfoxide (DMSO)/$H_2O$ at a ratio of 1/4 (*v/v*) and 0.05% (4 mM) calcein-AM in anhydrous dimethyl sulfoxide. The cell nuclei were counterstained with 4′,6-diamidino-2-phenylindole (DAPI) for 30 min at 37 °C and 5% $CO_2$. After that, the incubated cells were washed with PBS 3 times and then imaged using a BX53 digital upright microscope (Olympus, Tokyo, Japan) with a 10× objective lens. The experiment was repeated 5 times under the same conditions.

### 2.6. Sodium Dodecyl Sulfate–Polyacrylamide Gel Electrophoresis and Western Blot Analysis

Proteins were extracted from whole cells by lysis in radioimmunoprecipitation assay (RIPA) lysis buffer (Sigma–Aldrich, St. Louis, MO, USA) containing complete protease inhibitors. The total protein concentration was determined using a bicinchoninic acid (BCA) assay kit (Thermo Fisher, Waltham, MA, USA). The total protein (20 μg) was subjected to sodium dodecyl sulfate–polyacrylamide gel electrophoresis (SDS–PAGE) on a 4% stacking gel for 20 min at 80 V and an 8–12% separating gel for 90 min at 110 V. The proteins were transferred to polyvinylidene fluoride (PVDF) membranes (Millipore, St. Louis, MO, USA) using a Trans-Blot Turbo transfer system (Bio–Rad Laboratories Inc., Hercules, CA, USA) for 150 min at 200 mA. The membranes were blocked with blocking buffer that contained 5% skimmed milk in tris-buffered saline (TBS)-T buffer (10 mM HCl, pH 7.5, 100 mM NaCl, and 0.1% Tween-20) for 1.5 h at room temperature and then incubated with primary antibodies in 5% skimmed milk overnight at 4 °C. The experiment was repeated 5 times under the same conditions. The primary antibodies were as follows: anti-protein-disulfide isomerase (PDI; 1:500, Cell Signaling Technology, MA, USA), anti-C/EBP-homologous protein (CHOP; 1: 1000, Novus, CO, USA), anti-activating transcription factor 6 (ATF6; 1:500, Abcam, Cambridge, UK), anti-tumor growth factor-β (TGF-β; 1:500, Abcam, Cambridge, UK), anti-protein kinase RNA (PKR)-like endoplasmic reticulum kinase (PERK; 1:1000, Cell Signaling Technology, MA, USA), anti-glucose-regulated protein 78 (GRP78; 1:1000, Abcam, Cambridge, UK), anti-Janus kinase 2 (JAK2; 1:1000, Abcam, Cambridge, UK), anti-caspase-3 (1:1000, Santa Cruz, TX, USA), and anti-β-actin (1:1000, Abcam, Cambridge, UK). Then, the membranes were incubated with horseradish peroxidase (HRP)-conjugated secondary antibodies at room temperature for 1 h. Proteins were detected using chemiluminescence reagents (Thermo Fisher, Waltham, MA, USA), and the band densities were quantified with a ChemiDoc XRS system (Bio–Rad Laboratories Inc., Hercules, CA, USA).

### 2.7. Statistical Analysis

The data are presented as the mean ± standard deviation (SD). Comparisons between the groups were performed using two-way analysis of variance (ANOVA) with a Tukey–Kramer multiple comparison post hoc test using GraphPad InStat 3.05 software (GraphPad Software, La Jolla, CA, USA). Significance was defined as $p < 0.05$ compared with the control group. ImageJ (GitHub Inc., San Francisco, CA, USA) was used for image analysis and preparation.

## 3. Results

### 3.1. Quick Release of Erythropoietin from Erythropoietin Nanobots

The effects of one hour of preconditioning sonication on EPO release from the ENBs were determined at different time points up to 24 h, as shown in Figure 1. During the observation time, one hour of preconditioning sonication caused three times more release than no sonication (39.2% versus 8.4% at 4 h, 61.7% versus 21.2% at 12 h, and 84.1% versus 25.1% at 24 h, respectively). After 24 h, >85% of the EPO was released from the ENBs preconditioned with sonication.

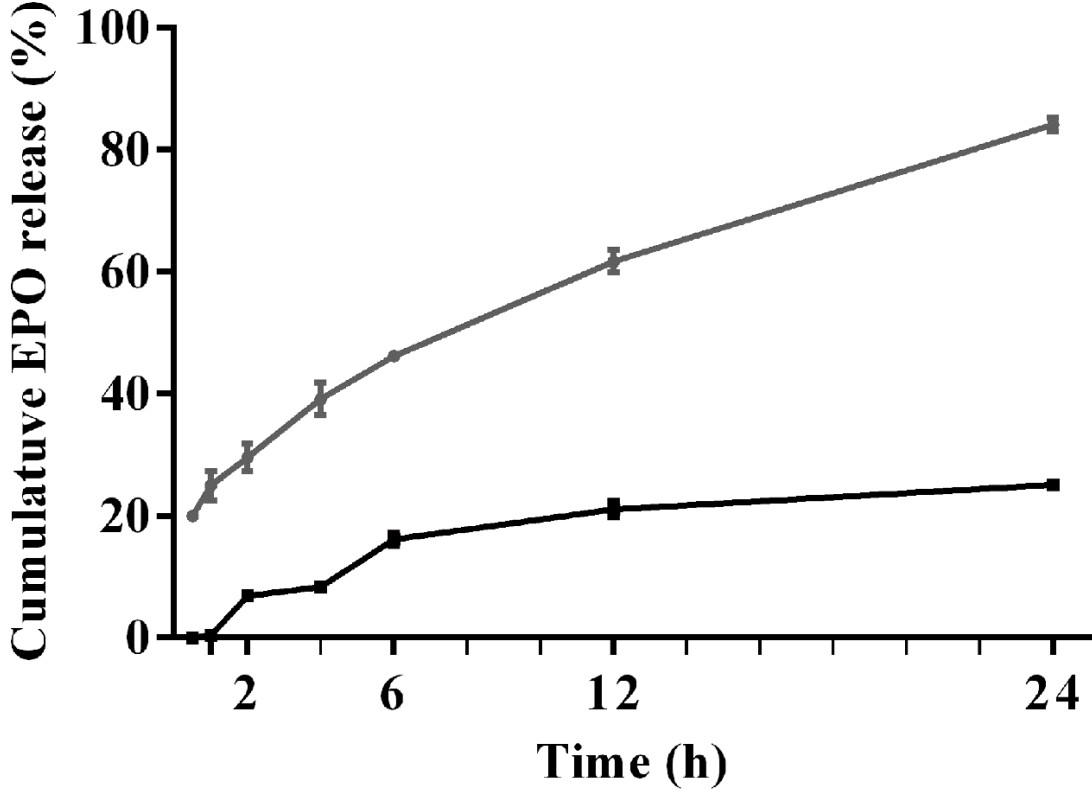

**Figure 1.** Cumulative release ratios of erythropoietin from the erythropoietin nanobots. The data are shown as the mean ± SD (error bars).

### 3.2. Neuronal Cell Viability after Erythropoietin Nanobot Treatment and Erythropoietin Treatment

Figure 2 shows the comparison of SH-SY5Y neuronal cell viability among the control (cell-only), TG, TG + EPO, and TG + ENB groups. Compared with the cell-only group, the TG group showed a continuous decrease in the number of live cells over time until 24 h. However, TG + EPO cotreatment resulted in more live cells than TG treatment, although there were fewer live cells in the TG + EPO group than in the cell-only control group. The TG + ENB and the TG + EPO groups showed similar numbers of live cells.

Figure 3 shows the quantitative results of the MTS cell viability assay. Immediately after treatment, all groups had similar viability. Two hours later, the cells in the TG group started to show considerable death with 73.5% viability at 2 h; the viability decreased to 15.2% by 24 h. However, the TG + EPO group and the TG + ENB group exhibited similar viabilities at all measured time points (93.5% and 89.1% at 2 h, respectively; both were over 65.0% at 24 h).

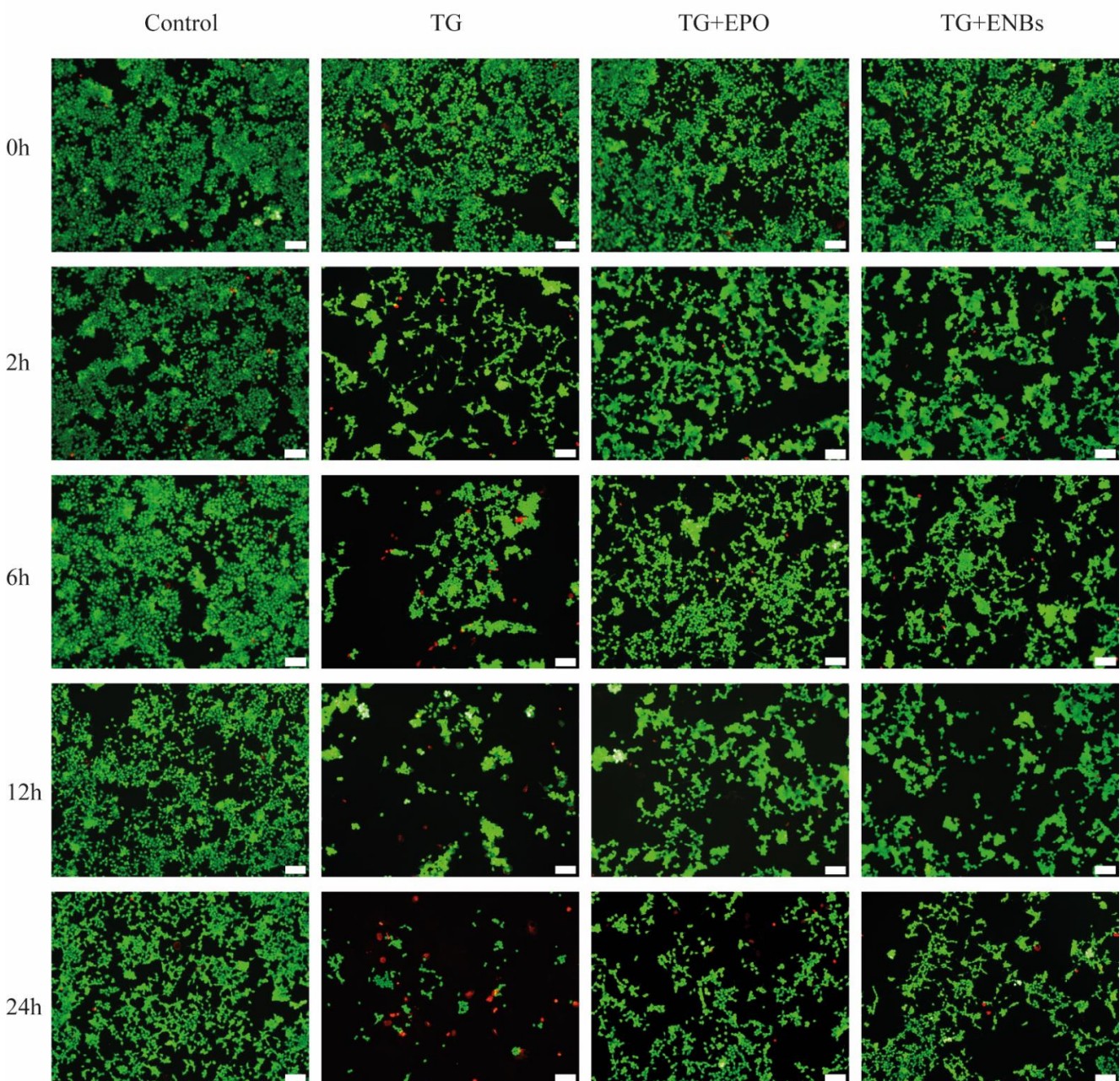

**Figure 2.** Live/dead fluorescence images of SH-SY5Y cells over 24 h. The control group in this experiment was the cell-only group. TG indicates thapsigargin treatment, TG + EPO indicates thapsigargin and EPO cotreatment, and TG + ENBs indicates thapsigargin and EPO nanobot cotreatment. Green represents live cells, while red represents dead cells. The scale bars are 100 µm.

### 3.3. Erythropoietin Receptor Signaling Pathway Activation

JAK2 expression was augmented at each point in the TG + EPO and TG + ENB groups, albeit to a lesser extent in the TG + ENB group (Figure 4). The highest level of JAK2 expression in the TG + EPO cotreatment group occurred at 6 h. In contrast, the highest expression in the TG + ENB group was exhibited at 12 h, showing a delay compared to the peak in the TG + EPO group. However, both treatments showed nearly the same level at 24 h, similar to the quantitative cell viability analysis results at 24 h (Figure 3).

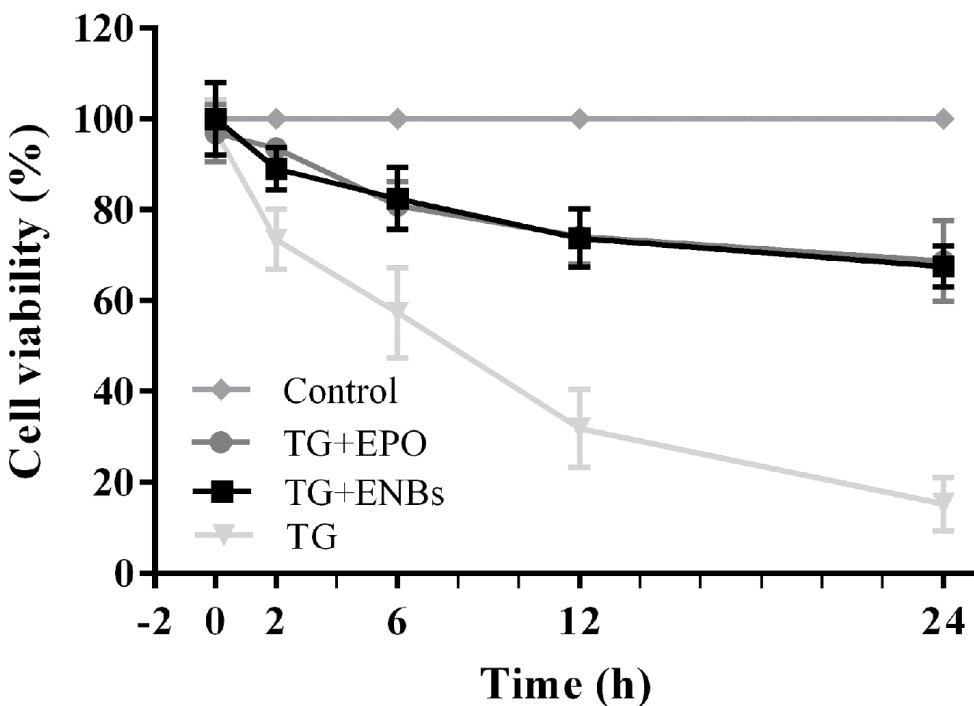

**Figure 3.** Viabilites of SH-SY5Y neuronal cells over 24 h. An MTS asssy was conducted. The control group was the cell-only group. The data are shown as the mean ± SD (error bars).

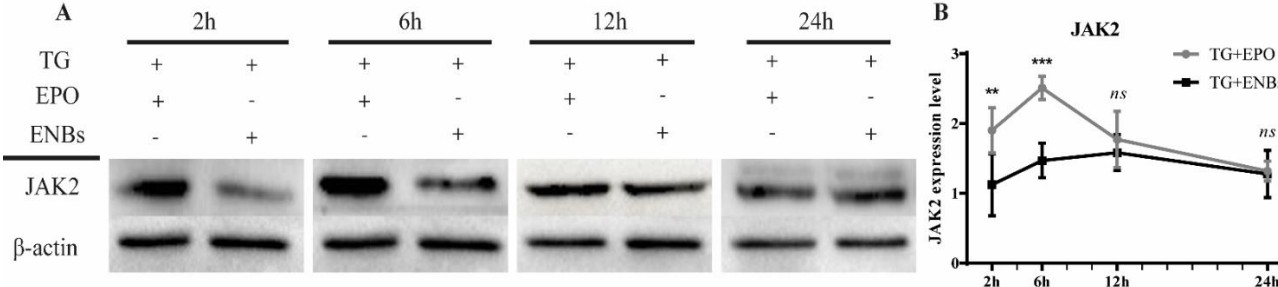

**Figure 4.** Janus kinase 2 expression (**A**) determined by Western blot analysis and (**B**) quantified over 24 h. β-Actin was used as a loading control. TG indicates thapsigargin treatment, TG + EPO indicates thapsigargin and EPO cotreatment, and TG + ENBs indicates thapsigargin and EPO nanobot cotreatment. JAK2, Janus kinase 2. ** $p < 0.01$, *** $p < 0.001$, and ns, not significant versus TG + EPO cotreatment.

### 3.4. Proapoptotic Signaling Pathway Deactivation

The TG + ENB and TG + EPO cotreatments decreased PDI and GRP78 expression levels over time, as shown in Figure 5B,C. Additionally, PERK and CHOP expression peaked at 6 h, unlike ATF6 expression, which peaked at 2 h; there was also a downward tendency by 24 h, similar to that for PDI and GRP78 expression (Figure 5D–F). Before 24 h, the TG + ENB and TG + EPO groups exhibited proapoptotic signaling pathway mediator expression in an analogous manner, although there was some time lag and a reduced degree of deactivation in the TG + ENB group; this result was similar to the results of EPO receptor signaling pathway activation (Figure 4). At 24 h, the expression levels of each proapoptotic signaling pathway biomolecular marker, except for ATF6, were nearly the same in both treatment groups, similar to the 24-h quantitative cell viability analysis results (Figure 3) and the EPO receptor signaling pathway activation results (Figure 4).

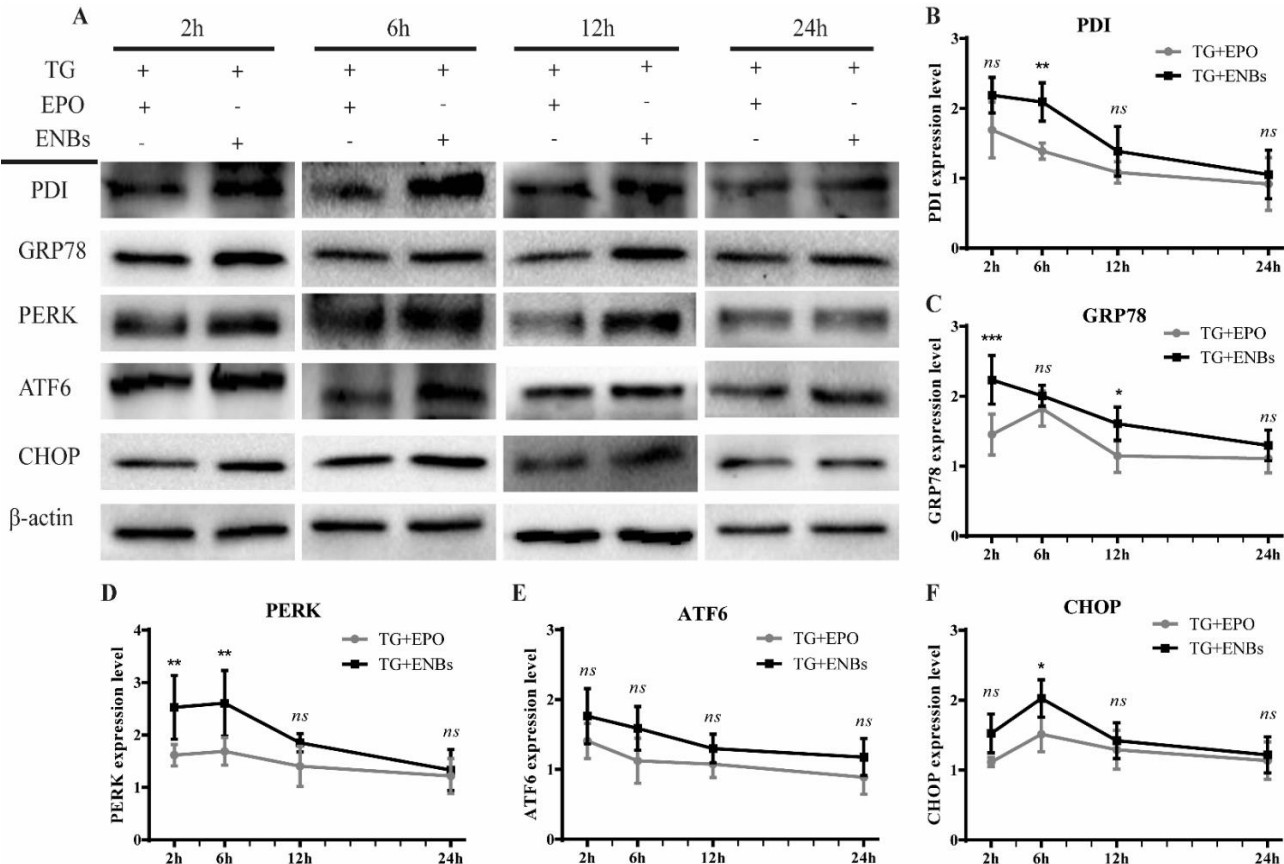

**Figure 5.** Endoplasmic reticulum stress-induced proapoptotic signaling pathway deactivation (**A**) determined by Western blot analysis and (**B–F**) quantified over 24 h. β-Actin was used as a loading control. TG indicates thapsigargin treatment, TG + EPO indicates thapsigargin and EPO cotreatment, and TG + ENBs indicates thapsigargin and EPO nanobot cotreatment. PDI, protein-disulfide isomerase; GRP78, glucose-regulated protein 78; PERK, protein kinase R-like endoplasmic reticulum kinase; ATF6, activating transcription factor-6; CHOP, C/EBP-homologous protein. * $p < 0.05$, ** $p < 0.01$, *** $p < 0.001$, and ns, not significant versus TG + EPO cotreatment.

### 3.5. Apoptotic Signaling Pathway Deactivation

Figure 6 shows that the TG + ENB and TG + EPO cotreatments reduced TGF-β expression over time until 24 h. From 2 h to 12 h of treatment, TG + EPO cotreatment diminished TGF-β expression more effectively than TG + ENB cotreatment (Figure 6A,B). Nevertheless, at 24 h, both treatment groups exhibited nearly the same level of TGF-β expression. The expression level of caspase-3 was reduced in both treatment groups, similar to the TGF-β expression pattern, except at 24 h, as shown in Figure 6A,C. The expression patterns of TGF-β and caspase-3 showed some time lag and a reduced degree of deactivation approaching 24 h, similar to the EPO receptor signaling pathway activation results (Figure 4) and the proapoptotic signaling pathway deactivation results (Figure 5).

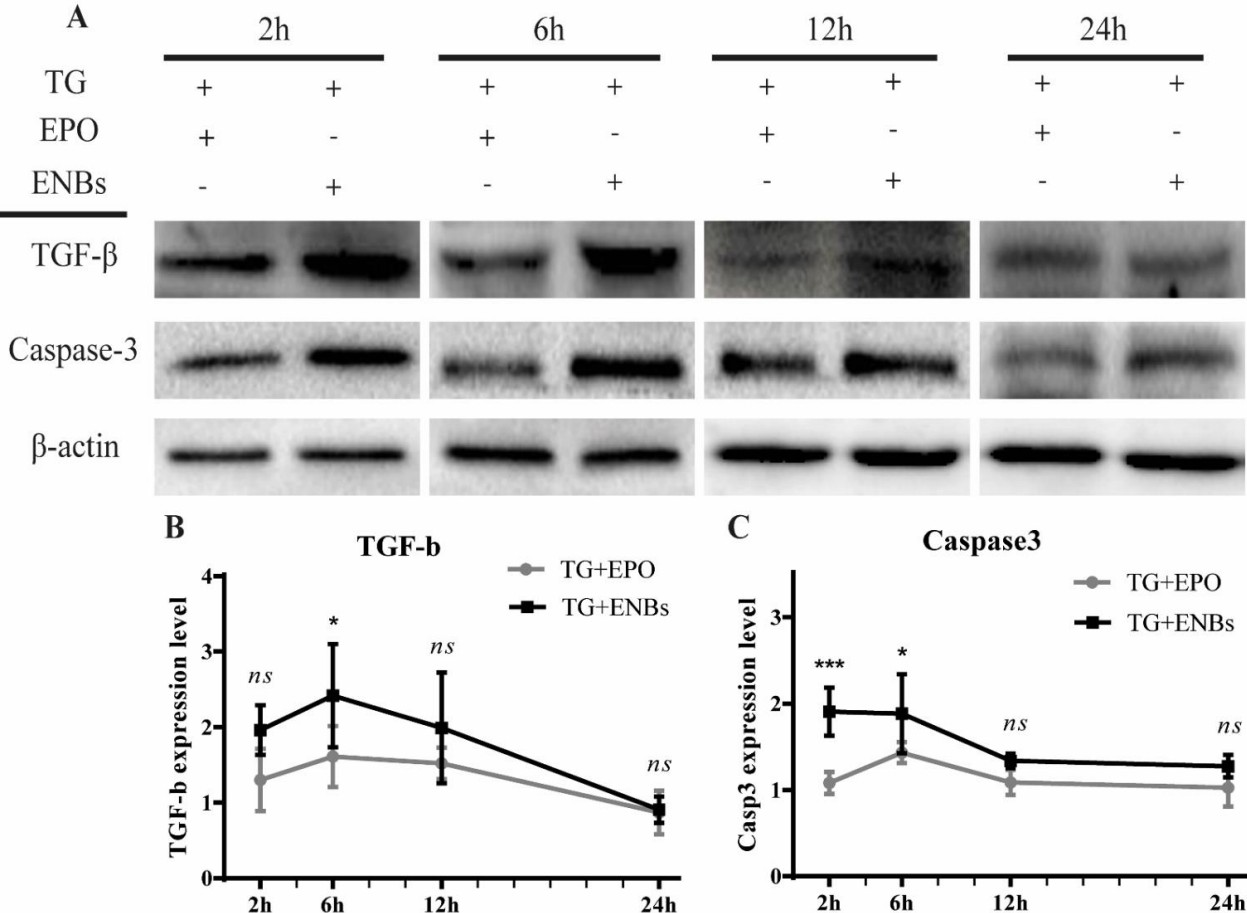

**Figure 6.** Endoplasmic reticulum stress-induced apoptotic signaling pathway deactivation (**A**) determinded by Western blot analysis and (**B**,**C**) quantified over 24 h. β-Actin was used as a loading control. TG indicates thapsigargin treatment, TG + EPO indicates thapsigargin and EPO cotreatment, and TG + ENBs indicates thapsigargin and EPO nanobot cotreatment. TGF-β, transforming growth factor-beta. * $p < 0.05$, *** $p < 0.001$, and ns, not significant versus TG + EPO cotreatment.

## 4. Discussion

To examine the feasibility of ENBs for the controlled release of EPO with neuroprotective bioequivalence to EPO as a therapeutic tool against CNS injury, the EPO release ratio after preconditioning sonication was evaluated over 24 h, and the cell viability and biomolecular marker expression were evaluated in vitro following ER stress induced by TG treatment. To manufacture the ENBs (i.e., EPO-MNPs encapsulated by an alginate polymer), 1.0 mL of a 1000 IU rhEPO solution was added to a mixture of 7.5 mg of sodium alginate and 150.0 mg of MNPs, and the mixed solution was sprayed using a nanospray drying technique. The key findings were as follows: (1) one-hour preconditioning sonication at a low frequency caused 61.7% of the EPO to be released from the ENBs at 12 h and 84.1% to be released at 24 h; (2) the SH-SY5Y neuronal cell survival ratios were nearly the same in both the TG + EPO and TG + ENB groups over 24 h (more than 65.0% viability at 24 h); (3) EPO receptor signaling pathway activation in the TG + ENB group showed a trend similar to that in the TG + EPO group, but to a lesser degree before 24 h, although these groups had nearly the same result at 24 h; (4) the proapoptic signaling pathway in the TG + ENB group presented a trend similar to that above, with a slight reduction before 24 h and almost the same level at 24 h compared to that in the TG + EPO group; (5) the apoptotic signaling pathway again showed a similar trend, as the TG + ENB group showed a slightly lower level before 24 h that became very similar to that in the TG + EPO group at 24 h.

One-hour of preconditioning sonication at 50–60 kHz caused a three-fold increase in EPO release from the encapsulating biodegradable alginate polymer compared to the release from the samples without preconditioning sonication (84.1% versus 25.1% at 24 h). This result is similar to the findings in Kost et al.'s report [19], in which 15 min of sonication at 75 kHz increased the rate of *p*-nitroaniline release from a biodegradable glycolide polymer by up to 20-fold, and in Rapoport et al.'s report [33], in which one hour of sonication at 20–70 kHz led to a greater than 10-fold increase in doxorubicin release from a biodegradable diethylacrylamide polymer. Similar to the case for unsonicated drug formulations encapsulated by biodegradable polymers, such as polylactic-co-glycolic acid, the drug release process relies solely on a diffusion mechanism [33,34]. However, preemptive or on-demand sonication can degrade sodium alginate polymers as well as other biodegradable polymers by disrupting the cross-links, which was expected to increase the release rate of EPO in the current research [19,20]. Similar to delivery studies with other therapeutics, most EPO delivery trials have focused on the sustained release of loaded EPO over a long period of time in peripheral nervous system injury models to guarantee a stable blood concentration on the concentration-time curve [35,36]. However, EPO shows a limited therapeutic time window (within 6–8 h) in in vitro models and in vivo small animal models of CNS injury [5–7,37]. Therefore, in clinical situations, the delivery and release of EPO should be as fast as possible for effective neuroprotective actions instead of sustained long-term release. This is the reason why fast breakdown of the alginate polymer encapsulating material is important. Furthermore, low-frequency (28 kHz) ultrasound more efficiently releases loaded drugs from nanobots than high-frequency (1 MHz) ultrasound [25], so low-frequency ultrasound should be chosen when fast breakdown of the polymer is required for a high release rate, as in the current study. In addition, sonication-induced polymer breakdown is not permanent; the cross-links in polymers, including alginate, can reform and quickly re-encapsulate the drugs during re-exposure to physiological fluids [20,21]. Thus, ENB breakdown should be reevaluated to determine whether there is actual binding between EPO and its receptors followed by downstream activation of the neuroprotective cascade from a bioequivalence point of view.

Regarding neuronal cell viability following TG treatment, the live/dead fluorescence images and MTS assay results showed that the TG + ENB cotreatment with preconditioning sonication and the TG + EPO-cotreatment gave similar numbers of live cells and similar cell viabilities (both more than 65.0%) as at 24 h. The neuroprotective actions of EPO have been extensively reported in a variety of in vitro and in vivo brain and spinal cord injury models, including chemical and mechanical models [3,38–41]. TG is known to induce ER stress by irreversible inhibition of the sarco-/endoplasmic reticulum $Ca^{2+}$-ATPase, finally leading to apoptosis [27]. Among various chemical neurotoxic agents, such as glutamate, kainate, and alpha-amino-3-hydroxy-5-methyl-4-isoxazole propionate (AMPA) [42–44], TG is currently the most commonly applied cytotoxic agent. In addition, SH-SY5Y cells are the most commonly used human-derived neuronal cell line in neurological studies [26]. Similar to other in vitro EPO experiments in neuronal cell injury models induced by chemical toxins (such as motor neuron injury induced by glutamate and cortical cell injury induced by a histone deacetylase inhibitor [45,46]), the current experiment revealed that ENBs exerted neuroprotective actions on cell viability similar to those of EPO after sonication-preconditioned autonomous breakdown in the TG-induced in vitro SH-SY5Y neuronal cell injury model.

To trigger the neuroprotective actions of EPO, the extrahematopoietic EPO receptor signaling pathway should be activated, which subsequently activates downstream mediators including the JAK2/signal transducer and activator of transcription 5 (STAT5) and phosphoinositide-3-kinase (PI3K)/protein kinase B (AKT) [5,37,47,48]. In the current study, compared with EPO, ENBs with preconditioning sonication showed delayed and reduced induction of JAK2 expression against TG-induced cytotoxicity before 24 h. This finding might have been caused by the greater time lag for alginate polymer breakdown and subsequent greater EPO release from ENBs than expected [19]. However, at 24 h, nearly

the same expression of JAK2 was found, indicating that at least 84.1% breakdown of ENBs is required for full neuroprotective execution. Moreover, in Hong et al.'s study, the time to peak JAK2 expression in the in vitro CNS injury models occurred after approximately 0 to 2 h of treatment [5], but this peak was observed at 6 h in the current study. However, different injury methods were applied (a combination of scratching with a needle tip and kainate treatment versus TG treatment). Either chemical or mechanical agents cause weaker injury modeling than a combination of the two [49]. Furthermore, kainate has 30 times greater neurotoxicity than glutamate, although no neurotoxicity comparison between kainate and TG has been reported [42]. Therefore, TG-induced injury might not have provoked sufficient injury compared with what was expected in the current trial.

ER stress-induced apoptosis can be measured with proapoptotic biomolecular markers. TG treatment provokes the accumulation of misfolded proteins, leading to ER stress [50,51]. Prolonged ER stress results in PERK signaling pathway activation, which initiates the unfolded protein response in the ER, leading to the promotion of a proadaptive/adaptive signaling pathway via global protein synthesis suppression and ATF6 activation [50,52–55]. Similar to the process in the adaptive response, the ER-resident chaperone binding protein GRP78 refolds the unfolded proteins in the ER [56,57], and PDI promotes oxidative protein folding through disulfide bond formation [58]. In contrast, CHOP, a downstream marker of the unfolded protein response, is an important proapoptotic transcription factor leading to programmed cell death [53,54,59]. In the current study, the TG + EPO group showed downward trends in the expression levels of proapoptotic molecular markers (PERK, ATF6, GRP78, PDI, and CHOP) over 24 h. As expected, the TG + ENB group with preconditioning sonication exhibited similar downward tendencies, but tendencies before 24 h were milder than those in the TG + EPO group. However, the expression level of each proapoptotic biomolecular marker was nearly the same in both groups at 24 h with the exception of ATF6.

Regarding ER stress-triggered apoptosis downstream, increased $Ca^{2+}$ leakage into the cytoplasm mediated by TG activates the TGF-β/Smad3 pathway [60,61], which induces intrinsic apoptosis in the affected cells [62,63]. In addition, activated caspase-3 (a cysteine protease) triggers DNA fragmentation in cells undergoing apoptosis [27,50]. Similar to the results of the aforementioned proapoptotic molecular marker assay, the TG + ENB group with preconditioning sonication presented similar downward trends in TGF-β and caspase-3 expression, but the trends were milder than those in the TG + EPO group before 24 h. The expression in these groups reached nearly the same level (TGF-β) or a similar level (caspase-3) at 24 h.

*Limitations and Future Outlook*

ENBs with preconditioning sonication showed a similar degree of neuroprotection as EPO over 24 h with respect to neuronal cell viability, EPO receptor activation, and proapoptotic and apoptotic pathway deactivation. However, regarding the activation/deactivation of the EPO receptor and proapoptotic and apoptotic pathways, neither TG + EPO nor TG + ENB cotreatment were compared to TG-only treatment (the negative control) in terms of the relative neuroprotective potency; thus, it is not certain how the neuroprotective effects of ENBs were substantially different after TG treatment. Moreover, although the ENBs activated the EPO receptor and deactivated the proapoptotic and apoptotic pathways to a lesser extent than EPO at 12 h and to nearly the same degree as EPO at 24 h, the neuroprotective actions of ENBs (the controlled release of EPO from ENBs by preconditioning sonication [50–60 kHz] for 1 h was most effective until 24 h after treatment), which have a therapeutic time window different from that of EPO in vitro/in vivo (6–8 h after insult) [5–7], raise concerns over their sufficiency to meet the clinical requirements. However, considering that a single systemic circulation cycle usually takes no more than 5 min, the achievement of enhanced and fast EPO release from the ENBs by adjustment of the preconditioning sonication conditions (for instance, a much longer duration or a lower frequency than the parameters in the current study) may ameliorate the therapeutic time window discrepancy (24 h versus 6 to 8 h), which would provide the current ENBs with approximately

96 chances for delivery to the injured CNS area over 8 h. However, if the breakdown of ENBs starts too early, before the ENBs arrive at the injured CNS area, the exposed MNPs could be engulfed by the mononuclear phagocytic system [64,65], the MNP-mediated targeted delivery capabilities, such as magnetic navigation [66], self-propulsion [67], and propulsion by tagged bacteria [68], may be weakened or lost. As alternatives to preserve the targeted delivery capabilities, the breakdown of ENBs may be postponed with the aid of in situ strategies, such as on-demand sonication [20], thermal conversion (via inverse piezoelectricity) [69], or the application of a high-frequency alternating magnetic field [70] after ENB arrival at the injured CNS area.

## 5. Conclusions

The well-known neuroprotective effects of EPO in CNS injury [1–3] have a limited therapeutic time window [5–9], heterogeneity of extrahematopoietic receptors [12], and phylogenetic differences; thus, the clinical application of EPO remains challenging. Targeted delivery using MNPs under magnetic navigation, which has been widely studied in cancer models [15–17], can be introduced as a solution. To prove the feasibility of ENBs (alginate-encapsulated EPO-MNPs manufactured using the nanospray drying technique) as potential therapeutic tools against CNS injuries, the authors evaluated the release controllability and neuroprotective bioequivalence of ENBs compared with EPO in TG-treated SH-SY5Y neuronal cells in vitro. ENBs subjected to one hour of preconditioning sonication (50–60 kHz) showed 84.1% EPO release by 24 h and resulted in a neuronal cell viability equal to that after treatment with EPO only over 24 h (more than 65.0%). Additionally, similar downward curves in the expression levels of biomolecular markers (JAK2, PDI, GRP78, PERK, ATF6, and CHOP), EPO receptor signaling pathway activation, and proapoptotic and apoptotic signaling pathway deactivation were observed; these changes were all attenuated after treatment with ENBs before 24 h, but the levels at 24 h were nearly the same between the ENBs and EPO. ENBs showed in vitro controllability of EPO release and in vitro feasibility for neuroprotection. These findings may lay a foundation for the potential in vivo use of ENBs as a neuroprotective therapeutic to treat CNS injury through adjusted preconditioning sonication, which balances fast release within the therapeutic time window of EPO and early engulfment of EPO by the mononuclear phagocytic system. To fully achieve the neuroprotective bioequivalence of ENBs to EPO in vivo, no less than 84.1% breakdown of the ENBs is required.

**Author Contributions:** Conceptualization, T.H.L., C.T.N., K.-i.K. and C.H.H.; data curation, K.-i.K.; formal analysis, T.H.L., C.T.N., K.-i.K. and C.H.H.; funding acquisition, K.-i.K. and C.H.H.; investigation, T.H.L. and C.T.N.; methodology, T.H.L., K.-i.K. and C.H.H.; project administration, T.H.L., C.T.N., K.-i.K. and C.H.H.; supervision, C.H.H.; validation, C.H.H.; visualization, T.H.L. and C.T.N.; writing—original draft, T.H.L., C.T.N., K.-i.K. and C.H.H.; writing—review and editing, C.H.H. All authors have read and agreed to the published version of the manuscript.

**Funding:** This work was supported by the Practical Technology Development Medical Microrobot Program (RS&D Center for Practical Medical Microrobot Platform, HI19C0642) funded by the Ministry of Health and Welfare (Republic of Korea) and the Korea Health Industry Development Institute (Republic of Korea).

**Institutional Review Board Statement:** Not applicable.

**Informed Consent Statement:** Not applicable.

**Acknowledgments:** The authors sincerely thank Choong Reen Kim for consultation on ER stress modeling.

**Conflicts of Interest:** The authors declare that there are no conflicts of interest.

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
