# Peer review of "Erythropoietin Nanobots: Their Feasibility for the Controlled Release of Erythropoietin and Their Neuroprotective Bioequivalence in Central Nervous System Injury"

_applsci, doi:10.3390/app12073351_

Round 1

Reviewer 1 Report

In this manuscript, Le et al examined the effect of preconditioning sonication on the controlled release of erythropoietin from nanobots in vitro.

Major concerns:

  1. As the authors have mentioned in the Introduction, the therapeutic time window is very limited (6–8 hours after insult). However, the controlled release of EPO from ENBs by preconditioning sonication (50–60 kHz) for 1 hour is most effective until 24hrs after treatment --- is this sufficient to meet the clinical requirement?  
  2. Fig 4 also showed that Jak2 expression level in the ENB-treated cells is still significantly lower than in the EPO-treated cells after 6 hours, which again raised the same concern as in point 1 above.

Reviewer 2 Report

The manuscript titled “Erythropoietin Nanobots: Their Feasibility for the Controlled Release of Erythropoietin and Their Neuroprotective Bioequivalence in Central Nervous System Injury” was weel prepared. Information included into this paper are very interesting for readers and innovative.

The introduction is clearly written, but there is no clearly defined research aim.

The results are very well described and present in graph/photos. Discussion section is comprehensive. Additionally, the Authors included into manuscript “Limitations and future outlook”, what significantly increases the value of paper.

The manuscript should be prepared with template dedicated for Applied Sciences.
